# Inappropriate Sinus Tachycardia Following Viral Illness

Khalid Sawalha [1,*], Fuad Habash [2], Srikanth Vallurupalli [2] and Hakan Paydak [3]

1   Internal Medicine Division, White River Health System, Batesville, AR 72501, USA
2   Cardiology Division, University of Arkansas for Medical Sciences, Little Rock, AR 72205, USA;
    FHabash@uams.edu (F.H.); SVallurupalli@uams.edu (S.V.)
3   Electrophysiology Division, University of Arkansas for Medical Sciences, Little Rock, AR 72205, USA;
    HPaydak@uams.edu
*   Correspondence: ksawalha@aol.com; Tel.: +1-984-364-1158

**Abstract:** A 67-year-old female patient with a past medical history of menopause, migraines, and gastro-esophageal disease presented with palpitation, fatigue, and shortness of breath. One month prior to her presentation, she reported having flu-like symptoms. Her EKG showed sinus tachycardia with no other abnormality. Laboratory findings, along with imaging, showed normal results. The event monitor failed to detect any arrythmias. We report a case of inappropriate sinus tachycardia secondary to viral infection as a diagnosis of exclusion.

**Keywords:** inappropriate sinus tachycardia; viral infection; palpitations





## 1. Introduction

Inappropriate sinus tachycardia, also called chronic non-paroxysmal sinus tachycardia, is an unusual condition that occurs in individuals without apparent heart disease or other cause of sinus tachycardia, such as hyperthyroidism or fever, and is generally considered a diagnosis of exclusion [1–4]. Inappropriate sinus tachycardia is defined as a resting heart rate >100 beats per minute associated with highly symptomatic palpitations [5,6].

Commonly used criteria to define inappropriate sinus tachycardia include [7] P-wave axis and morphology similar to sinus rhythm, and a resting heart rate of 100 beats per minute or greater (with a mean heart rate >90 beats per minute over 24 h). Additionally, patients with inappropriate sinus tachycardia classically experience a drop in heart rate during sleep, as well as palpitations, presyncope, or both, related to the tachycardia. Very rarely do patients experience syncope. Exclusion of identifiable causes of sinus tachycardia is the method of diagnosis.

## 2. Case Presentation

A 67-year-old female patient with a past medical history of menopause, migraines, and gastro-esophageal disease presented with palpitation, fatigue, and shortness of breath. One month prior to her presentation, she reported having flu-like symptoms. Her home medications include estradiol 0.025 mg weekly, glucosamine/chondroitin 500–400 mg twice daily, omeprazole 50 mg once daily, and zolmitriptan 5 mg as needed. Physical examination was normal.

Laboratory results were as follows: WBC 8.64 K/uL (3.60–9.50), creatinine 0.8 mg/dL (0.4–1.0), TSH 1.69 uIU/mL (0.34–5.60), troponin < 0.03 ng/mL (≤0.4), D-dimer 387 ng/mL (<670), BNP < 10 pg/mL (≤106), CRP < 5.00 mg/L (≤10.00), procalcitonin 0.04 ng/mL (0.00–0.10); urine analysis negative; COVID-19 PCR infection negative; chest X-ray with no acute findings; echocardiograph with grade I diastolic dysfunction; and EF 60–65% (Figures 1 and 2). EKG showed sinus tachycardia with a heart rate of 101 bpm (Figure 3). Lung CTA was observed with no significant lung pathology, and the pulmonary function test was within normal limits.

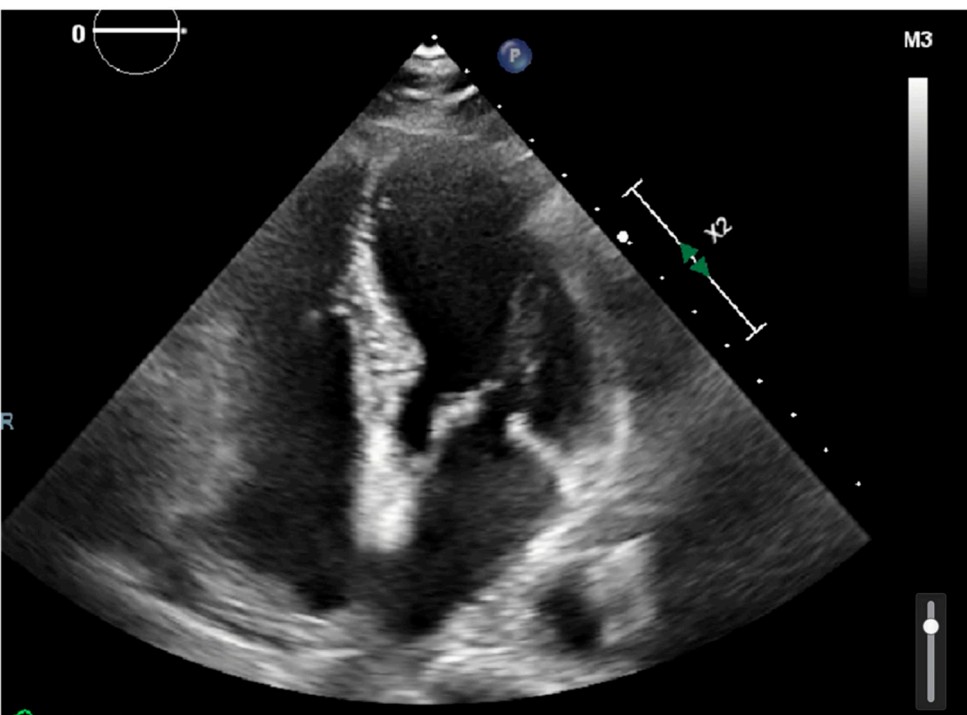

**Figure 1.** Echocardiography showing four-chamber view with no valvular or structural abnormalities.

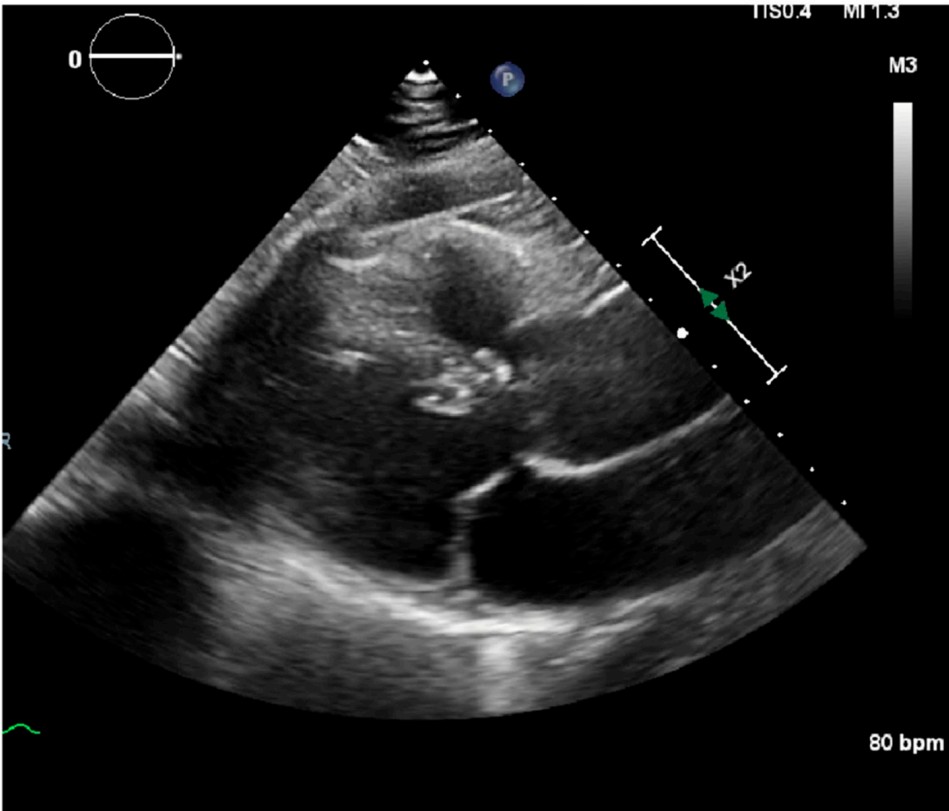

**Figure 2.** Echocardiography showing mitral valve and left ventricle with no pericardial effusion.

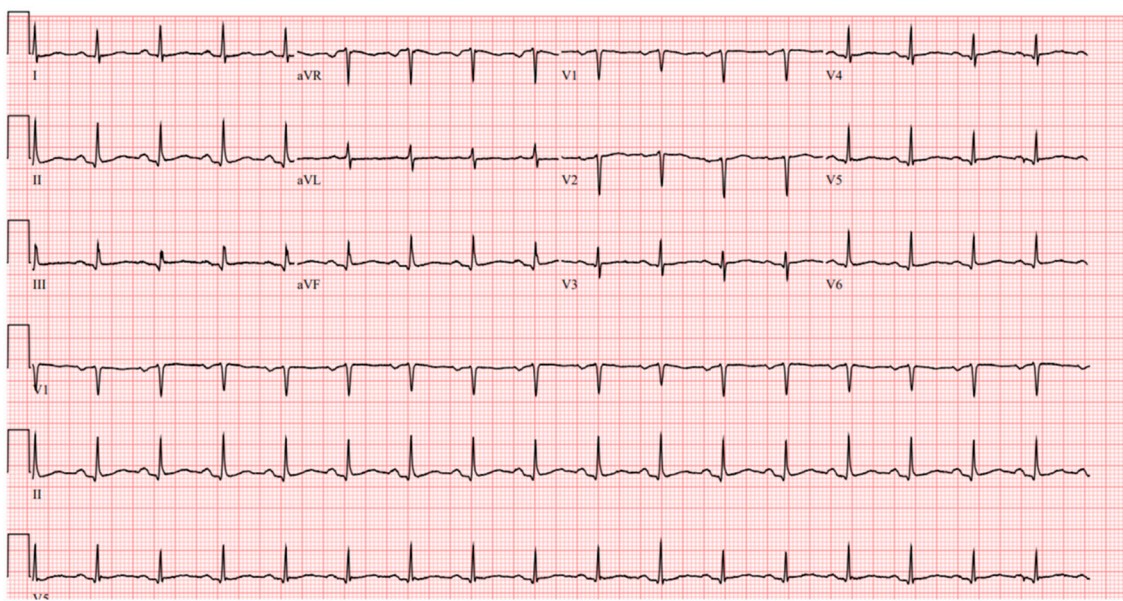

**Figure 3.** EKG showing sinus tachycardia with heart rate of 101 bpm.

The patient started on metoprolol succinate 25 mg daily and was set up with an event monitor for a two-week duration to rule out any arrythmias. She was advised to practice relaxation exercises, such as yoga and tai chi. At the one month follow-up, the event monitor showed no arrhythmias. Only one episode of asymptomatic premature ventricular complex was detected. The patient reported improvement with her symptoms, and, thus, she was diagnosed with inappropriate sinus tachycardia, as part of her previous viral illness, by exclusion.

## 3. Discussion

Most patients with inappropriate sinus tachycardia are young and female. Among a single-center cohort of 305 patients with inappropriate sinus tachycardia seen between 1998 and 2018, 92% were female, with an average age of 33 years [8]. Affected patients have an elevated resting heart rate and/or an exaggerated heart rate response to exercise that is out of proportion to their body's physiological needs; many patients have both.

Patients are invariably symptomatic; the presence of symptoms is an essential component of the definition. In contrast to sinus tachycardia occurring as a physiologic response, inappropriate sinus tachycardia can continue for months or years and may produce troublesome symptoms, most commonly palpitations. However, other common symptoms include chest discomfort, fatigue, dizziness, presyncope, syncope, and shortness of breath [8]. Most patients have resting heart rates of greater than 100 beats per minute and average heart rates on a 24 h Holter greater than 90 beats per minute, with no clear physiologic, pathologic, or pharmacologic trigger [4].

The pathophysiologic mechanism behind this disease is poorly understood and is thought to consist of intrinsic sinus node hyperactivity coupled with autonomic perturbations modulated by neurohormonal influences [4]. One study suggested that this tachycardia is related to a primary sinus node abnormality characterized by a high intrinsic heart rate, depressed efferent cardiovagal reflex, and beta-adrenergic hypersensitivity [2,4].

Inappropriate sinus tachycardia is an unusual condition of unknown etiology that occurs in individuals without apparent heart disease or other cause of sinus tachycardia. Before embarking on treatment, exclusion of secondary causes of sinus tachycardia is imperative, as it is a normal physiologic response to exercise and conditions in which catecholamine release is physiologically enhanced or, less commonly, in situations where the parasympathetic nervous system is withdrawn. Among the most common causes are

anxiety and pain. Patients with such presentation will often require simple reassurance with return of the sinus rate to the normal range [9].

Treatment of symptomatic inappropriate sinus tachycardia is challenging, often with suboptimal results. For patients with symptomatic inappropriate sinus tachycardia, a trial of beta blockade as the initial medical therapy with a daily dose of 25 to 50 mg long-acting metoprolol, with upward titration for adequate heart rate and symptom control classified as 2C, is suggested. For patients with persistently symptomatic inappropriate sinus tachycardia, using ivabradine (5 to 7.5 mg twice daily) as well as a class 2C classification, is suggested. For patients with persistent symptomatic inappropriate sinus tachycardia despite optimal pharmacologic therapy, radiofrequency catheter ablation may be attempted. However, postural orthostatic tachycardia syndrome must be excluded first since ablation may worsen symptoms in these patients.

## 4. Conclusions

Inappropriate sinus tachycardia is a diagnosis of exclusion after ruling out causes of sinus tachycardia. It can be seen as a physiological response after a viral illness, as seen in this case. Treatment is often challenging and with suboptimal results. However, long-acting b-blockers are recommended as initial medical therapy.

**Author Contributions:** K.S.—Writing and gathering date for the manuscript; F.H., S.V. and H.P.—Reviewing the manuscript. All authors have read and agreed to the published version of the manuscript.

**Funding:** This manuscript received no funding.

**Institutional Review Board Statement:** Not applicable.

**Informed Consent Statement:** Verbal Consent was obtained directly from the patient to publish this manuscript.

**Data Availability Statement:** Data available upon request.

**Conflicts of Interest:** The authors declare no conflict of interest.

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
