# Peer review of "Inappropriate Sinus Tachycardia Following Viral Illness"

_clinpract, doi:10.3390/clinpract11020032_

Round 1

Reviewer 1 Report

This case report focuses on inappropriate sinus tachycardia, which is a condition often misdiagnosed. It is important to recognize and treat it in order to improve the discomfort associated with this condition. Several aspects of the presented clinical case should be improved:

1) please add some figures such as echocardiography images that may demonstrate the absence of abnormal morphological myocardial alterations.

2) please discuss the role of anxiety both in the onset of inappropriate sinus tachycardia as well as in the progression of this and other tachyarritmias (Card Res Pract 2019 Feb 18;2019:1208505. doi: 10.1155/2019/1208505.)

3) why cardiac RM was not considered in order to exclude subclinical or undetected  myocarditis? 

Author Response

Dear reviewer 1,

Thank you for your comments and time, these are excellent comments and I have tried to reply as best I could below. If there are any leading questions, I’ll be happy to take them. Looking forward for your responses. 

  • please add some figures such as echocardiography images that may demonstrate the absence of abnormal morphological myocardial alterations.

Reply: Edits are made as requested.

  • please discuss the role of anxiety both in the onset of inappropriate sinus tachycardia as well as in the progression of this and other tachyarritmias (Card Res Pract 2019 Feb 18; 2019:1208505. doi: 10.1155/2019/1208505.)

Reply: Edits are made as requested and the reference was cited. Thank you. 

  • why cardiac RM was not considered in order to exclude subclinical or undetected myocarditis? 

Reply: Based on her clinical presentation and results including event monitor and echocardiogram and improvement of her symptoms with metoprolol it wasn’t done. However, if not improvement it was definitely considered.  

Reviewer 2 Report

clinpract 1168007 review

Inappropriate sinus tachycardia following viral illness

The authors reported a patient who eventually was diagnosed as inappropriate sinus tachycardia following viral illness, which was treated by a long acting beta-blocker.

  1. How was the effect of metoprolol in this patient?
  2. How about the mood disorder as a cause of sinus tachycardia in this patient?

Author Response

Dear reviewer 2,

Thank you for your comments and time, these are excellent comments and I have tried to reply as best I could below. If there are any leading questions, I’ll be happy to take them. Looking forward for your responses. 

  1. How was the effect of metoprolol in this patient?

Reply: Her symptoms have improved significantly on her follow up visit. She reported feeling better so the metoprolol was effective.

  1. How about the mood disorder as a cause of sinus tachycardia in this patient?

Reply: This is a good thought. However, our patient didn’t have any history of depression or mood disorders. Neither any clinical suspicion that points to it.

Round 2

Reviewer 1 Report

All the comments and suggestions have been exhaustively replied 

Reviewer 2 Report

There are no further comments.